# Automated Claim Detection for Fact-checking: A Case Study using Norwegian Pre-trained Language Models

**Ghazaal Sheikhi**
MediaFutures
University of Bergen

**Samia Touileb**
MediaFutures
University of Bergen

**Sohail Ahmed Khan**
MediaFutures
University of Bergen

## Abstract

We investigate to what extent pre-trained language models can be used for automated claim detection for fact-checking in a low resource setting. We explore this idea by fine-tuning four Norwegian pre-trained language models to perform the binary classification task of determining if a claim should be *discarded* or *upheld* to be further processed by human fact-checkers. We conduct a set of experiments to compare the performance of the language models, and provide a simple baseline model using SVM with tf-idf features. Since we are focusing on claim detection, the recall score for the *upheld* class is to be emphasized over other performance measures. Our experiments indicate that the language models are superior to the baseline system in terms of F1, while the baseline model results in the highest precision. However, the two Norwegian models, NorBERT2 and NB-BERT$_{large}$, give respectively superior F1 and recall values. We argue that large language models could be successfully employed to solve the automated claim detection problem. The choice of the model depends on the desired end-goal. Moreover, our error analysis shows that language models are generally less sensitive to the changes in claim length and source than the SVM model.

## 1 Introduction

With the growing concerns about misinformation, fact-checking has become an essential part of journalism. To mitigate the time and the human burden of fact-checking and to allow for more fact-checked articles, automated fact-checking (AFC) systems have been developed (Guo et al., 2022; Zeng et al., 2021; Lazarski et al., 2021). To approach automated fact-checking, three basic tasks are defined in the pipeline: claim detection, evidence retrieval, and claim verification. Claim detection refers to monitoring social media and political sources for identifying statements worth checking. The subsequent components retrieve reliable documents for debunking the detected claims and generate a verdict. Several tools have been developed to automate these tasks to meet the expectations of the human fact-checkers[1]. According to the studies on the user needs of fact-checkers, claim detection receives the highest preference among other AFC tools (Graves, 2018; Dierickx et al., 2022). Automated claim detection is a classification problem, where models are trained on sentences parsed from text documents and labelled by humans according to their check-worthiness (Hassan et al., 2017a).

In this work, we explore how well Norwegian pre-trained language models (LMs) perform on the task of automated claim detection. This is, to the best of our knowledge, the first attempt at automated claim detection for Norwegian using LMs. Fine-tuning LMs for the task of automated claim detection is not novel (Cheema et al., 2020; Zhuang et al., 2021; Shaar et al., 2021). However, this has never been done on Norwegian, and we believe that our insights into which errors these models do compared to simple baselines is a valuable contribution. Our research questions are:

- How well do Norwegian LMs perform on the task of automated claim detection compared to a simple SVM baseline?

- Which aspects of claim detection do these LMs still struggle with?

---

[1]https://www.rand.org/research/projects/truth-decay/fighting-disinformation/search.html

To address these questions, we first fine-tune each model on a small dataset from a Norwegian non-profit fact-checking organization, comprising claims manually annotated with labels reflecting their check-worthiness. Then we manually analyse the misclassifications of each model and provide an error analysis.

We believe that the contributions of this work have important societal implications. The case we study here sheds lights on the future directions of claim detection tools for fact-checking based on pre-trained language models for low to medium resourced languages. This would contribute to the fight against dis/misinformation by scaling and speeding up the fact-checking process.

The rest of the paper is organized as follows. In Section 2 we give an overview of previous work on automated claim detection. Section 3 describes the dataset and our experimental setup. We present and discuss our results and provide an error analysis in Section 4. Finally, we summarize our main findings, and discuss possible future works in Section 5.

## 2 Background

Automated claim detection for fact-checking does not have a long history, but it has turned to be one of the attractive fields of research in NLP (Hassan et al., 2015; Gencheva et al., 2017; Beltrán et al., 2021; Cheema et al., 2020; Shaar et al., 2021). One of the first studies on claim detection for AFC is initiated as part of the ClaimBuster project Hassan et al. (2017b). Their initial claim detection system was based on a set of features (sentiment, word count, part of speech (PoS) tags and named entities (NE)) followed by a feature selection and a traditional classifier namely Naive Bayes, SVM, and Random Forest(Hassan et al., 2015). Claim detection has also been addressed in languages other than English. ClaimRank is a claim detection system that supports both Arabic and English (Gencheva et al., 2017). A comprehensive set of features such as tf-idf, assertiveness, subjectivity, word embeddings are added to the ClaimBuster features and are fed to a two-layered neural network classifier (Gencheva et al., 2017).

In recent years, employment of pre-trained language models (LMs) in automated claim detection has been considered by numerous researchers (Cheema et al., 2020; Shaar et al., 2021; Beltrán et al., 2021). Several instances of these works are presented in the check-worthiness detection sub-tasks in CLEF CheckThat! editions (introduced in 2018 and ongoing) (Shaar et al., 2021; Nakov et al., 2022). CheckThat! provides data sets in different languages (English, Turkish, Arabic, Bulgarian, and Spanish) for the claim detection task on Twitter and political debates. The teams participating in this task have proposed classifier models mostly based on LMs. For instance, the top-ranked teams in CheckThat! 2020 used BERT (Devlin et al., 2019) and RoBERTa (Zhuang et al., 2021) with enhanced generalization capability (Williams et al., 2020) to detect check-worthy Tweets. For the task of detecting claims in political debates, the baseline BiLSTM (Schuster and Paliwal, 1997) model with GloVe embedding outperforms the LM-based systems (Martinez-Rico et al., 2020). ClaimHunter (Beltrán et al., 2021) is another BERT-based claim detection system that leverages XLM-RoBERTa [2](Conneau et al., 2020), a multilingual version of RoBERTa. It has been proved that the proposed model is superior to the classical baseline models NNLM+LR (Neural-Net Language Models embedding+Logistic Regression) and tf-idf+SVM.

To deal with the problem of small training data for LMs, data augmentation is employed. Claim detection from Twitter has been approached by generating synthetic check-worthy claims with lexical substitutions using BERT-based embeddings (Shaar et al., 2021). This approach improves the performance of BERT (Devlin et al., 2019) and RoBERTa (Liu et al., 2019) classification models (Shaar et al., 2021). It has also been shown that the BERTweet (Nguyen et al., 2020) model, fine-tuned on claims normalized and augmented by substitutions using WordNet, surpasses a reference n-gram model (Shaar et al., 2021).

## 3 Experiments

### 3.1 Data set

The data set is provided to us by Faktisk.no AS [3], a non-profit fact-checking organization and independent newsroom in Norway. Faktisk is jointly owned by several prominent Norwegian media houses, including VG, Dagbladet, NRK, TV2, Polaris Media, and Amedia. As per the company's articles of association, it operates under the overar-

---

[2] https://huggingface.co/xlm-roberta-large
[3] https://www.faktisk.no/om-oss

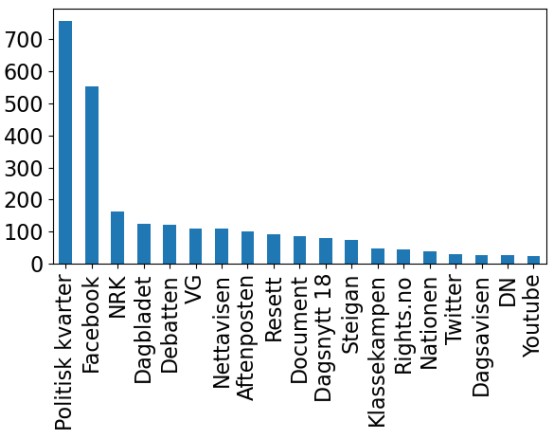

Figure 1: Most frequent sources of claims in our dataset provided by *Faktisk*, the non-profit fact-checking organization and independent newsroom in Norway.

ching ethical guidelines for the Norwegian press, as stipulated in the Vær Varsom poster [4]. To ensure its editorial and organizational independence, Faktisk.no adheres to the provisions of the Media Responsibility Act [5] and its articles of association. This ensures the editor's autonomy from the influence of the owners and other interested parties with interests in Faktisk's affairs. Thus, the funding news organizations of Faktisk and this project, being a source of some of the claims in the dataset should not raise concern about the independence of this research.

The data set comprises 4885 claims in Norwegian collected from social debates and public discourses from 04.03.2018 to 20.05.2022. Each claim in the dataset is provided with its respective source. These cover a selection of Norwegian newspapers (Dagbladet, VG, Nettavisen, Aftenposten, Klassekampen, Nationen, Dagsavisen, DN), alternative news outlets (Resett, Steigan, Document), think tank (Rights.no), the Norwegian Broadcasting Corporation (NRK), social media (Facebook, Twitter, YouTube), and TV/Radio (news) shows (Dagsnytt18, Politisk kvarter, Debatten). The alternative news outlets and the think tank are generally considered radical and controversial. The distribution of the occurrence of these sources can be seen in Figure 1.

A label is assigned to each claim, which refers

[4] http://presse.no/pfu/etiske-regler/vaer-varsom-plakaten/
[5] https://lovdata.no/dokument/NL/lov/2020-05-29-59

to the actions taken by human fact-checkers. This data set has been labelled as part of the daily routine in the organization *Faktisk.no* and is neither hand-crafted nor crowd-sourced for training LMs. Thus, it resembles a real world problem. The data set labels are {*Discarded, Checked and rejected, Pre-checked, Published, Suspended, Checked, Facebook*}. After removing the missing values, the rare samples with label 'Facebook' (only nine claims), and the short claims with less than five words, we end up with 4116 claims across six different labels. These labels are produced during the fact-checking procedure. According to Faktisk, a claim must be based on verifiable information and should not be normative or a prediction of the future. For a claim to be considered for fact-checking, it must be supported by verifiable information and should not involve predictions or normative statements about the future. Additionally, the claim should have a certain degree of controversy and relevance to a majority of people. Less relevant claims may be fact-checked if they possess good entertainment value. Once a claim is selected, an attempt is made to contact the sender to verify the claim and its surrounding context. In cases where the sender is unknown, the origin and context of the claim are used as the starting point for the fact-checking process.

For our purposes, we aim to focus on class labels specified as whether a claim is worth being considered for further processing or if it should be discarded. We therefore define a binary classification task with the labels *Discarded* and *Upheld*; where the *Discarded* class refers to claims with the same label (Discarded) in the data set, and the *Upheld* class includes the claims originally labelled as *Pre-checked and rejected, Pre-checking, Published, Suspended, or Checking*. A brief explanation of these labels as well as the mapping of the original labels to the binary class labels is given in Table 1. The number of claims in each category is also presented. There are 2810 claims in the first class and 1306 claims in the second class. The average and the maximum length of claims in these samples are equal to 16 and 107 words, respectively.

### 3.2 Experimental setup

**Pre-trained language models** We fine-tune four Norwegian LMs to perform the binary classification task of claim detection. Norwegian

| Class | Data Set Label | Description | #Claims |
|---|---|---|---|
| Discarded | Discarded | The claim has simply been discarded, there is no need for further investigation. | 2810 |
| Upheld | Pre-checked and rejected | Some preliminary work has been done to see if the claim is worth fact-checking, with a negative result. | 372 |
| | Pre-checking | Preliminary work to see if the claim is worth fact-checking has been started. | 336 |
| | Published | The fact-check about the claim has been published. | 297 |
| | Suspended | The claim will be taken up for consideration at a later time, and pre-checking or fact-checking will start then. | 194 |
| | Checking | A fact-check about the claim is in progress. | 107 |

Table 1: Distribution of claims across class labels and related labels in our dataset.

has two official written standards: Bokmål and Nynorsk, and the four models are trained on data in both written forms. These are:

- **NorBERT** (Kutuzov et al., 2021): trained on the Norwegian newspaper corpus[6], and Norwegian Wikipedia, with a vocabulary of about two billion word tokens.

- **NorBERT2**[7]: trained on the non-copyrighted subset of the Norwegian Colossal Corpus (NCC)[8] and the Norwegian subset of the C4 web-crawled corpus (Xue et al., 2021). The size of the vocabulary is about 15 billion word tokens.

- **NB-BERT**[base] (Kummervold et al., 2021): trained on the full NCC, and follows the architecture of the BERT cased multilingual model (Devlin et al., 2019). This model is bigger than the two previous ones, and comprises around 18.5 billion word tokens.

- **NB-BERT**[large][9]: trained on NCC, and follows the architecture of the BERT-large uncased model. This model is bigger and

trained on more data (from the same sources) than it's base-form NB-BERT[base].

**Training details** The baseline model is a SVM classifier with tf-idf features (Jones, 2004), implemented using the Scikit-learn library[10]. To split the data, stratified sampling based on the original data set labels is employed to ensure the distributions of the real world label noise is consistent among the splits. The ratio of the train, validation, and test sets is $70\% - 20\% - 10\%$ respectively. The validation set is employed to tune the hyperparameters of the model. To account for class imbalance, weighted F1 is used for scoring, which computes metrics for individual labels and determine their weighted average based on their respective support values. The hyperparameters of the best model are (C=100, gamma=0.1, kernel='rbf'). It should be noted that the preliminary experiments revealed that the baseline model performs extremely poor on the minority class, *Upheld*. To make a fair comparison between the baseline model and the BERT-based models, we have examined five different random states for splitting the data and chosen the one in favour of the baseline model. Furthermore, we ensured that the distribution of the length of claims in the test split is consistent with the whole data set (See Figure 3 (a)). The same split is used for fine-tuning the pre-trained LMs. The selected split results in the highest F1 for the *Upheld* class by the baseline

---

[6]https://www.nb.no/sprakbanken/ressurskatalog/oai-nb-no-sbr-4/
[7]https://huggingface.co/ltgoslo/norbert2
[8]https://github.com/NbAiLab/notram/blob/master/guides/corpus_description.md
[9]https://huggingface.co/NbAiLab/nb-bert-large

[10]https://scikit-learn.org/stable/

| Hyperparameter | Value |
|---|---|
| batch_size | 16 |
| init_lr | 2e-5 |
| end_lr | 0 |
| warmup_proportion | 0.1 |
| num_epochs | 5 |
| max_seq_length | 64 |

Table 2: Hyperparameter configuration of the four used Norwegian language models.

| Model | t (s) | P | R | F1 |
|---|---|---|---|---|
| **tf-idf+SVM** | 2 | **0.440** | 0.168 | 0.243 |
| **NorBERT** | 44 | 0.328 | 0.626 | 0.430 |
| **NorBERT2** | 45 | 0.401 | 0.588 | **0.477** |
| **NB-BERT**$_{base}$ | 48 | 0.358 | 0.336 | 0.345 |
| **NB-BERT**$_{large}$ | 103 | 0.320 | **0.740** | 0.447 |

Table 3: Training time and claim detection results for the used models, in terms of precision (P), recall (R), and F1.

model among the five examined random splits.

The claim detection models are fine-tuned using a TensorFlow-based model for sequence classification [11] from the HuggingFace `transformers` library [12]. Bert-based model transformer have a sequence classification head, i.e. a linear layer on top. We use the same train, validation, and test splits as the baseline model and the validation set is deployed to return the best model after five epochs. All experiments are repeated for five times and the best run in terms of F1 is reported. All models are fine-tuned with Adam optimizer (Kingma and Ba, 2014). The other hyperparameter configurations are identical for all the four models, and can be seen in Tabel 2.

# 4 Results and discussion

## 4.1 Classification performance

The performance of the classification models on the test data are measured in terms of precision, recall, and F1. The *Upheld* class is treated as the positive class. It should be noted that in automated claim detection, overlooked important claims have a higher cost than misclassified unim-

---

[11]`TFAutoModelForSequenceClassification`
[12]`https://huggingface.co/docs/transformers/index`

portant claims. In other words, the recall score of the *Upheld* class should be given particular emphasis.

Table 3 presents the results for the baseline model and the four fine-tuned language models. Metrics are computed for the positive class. The highest score in each column is shown in bold. For the case of precision, the baseline system outperforms the LMs, but recall and F1 are extremely poor. It is noticeable how all the four LMs are superior to the baseline system in terms of F1, with NorBERT2 standing on the top. Another significant reflection of the results is NB-BERT$_{large}$'s superior performance in terms of recall. The training time (in seconds) is also given in the table. We run the experiments on a PC with an AMD Ryzen 7 5800X 8 Core Processor, an Nvidia GeForce RTX-3080 GPU with 10 GB graphics memory and 32 GB of RAM. The largest model, NB-BERT$_{large}$, requires twice as much training time compared to the other three models.

## 4.2 Error analysis

To get insights on the errors made by our models, confusion matrices of the predictions are plotted in Figure 2. The horizontal and vertical axes refer to the predicted and true labels, respectively. If we focus on one of the classes in terms of precision-recall, the baseline model and NB-BERT$_{large}$ are the best models. These models appear to learn one of the classes better, having fewer errors on that class. For example, NB-BERT$_{large}$ has learnt to correctly classify more instances of the upheld class. But the fact that it also classifies a large proportion of the claims from the discarded class as upheld shows that it simply has overfitted on the upheld class. This observation seems to be true for the SVM model (overfitted to the majority class) as well, and to some extent can be said about NB-BERT$_{base}$.

NorBERT and NorBERT2 seem to actually learn a more decent representation of the label distribution. While NorBERT exhibits some similarities with the previous models, by mostly classifying claims as one class rather than the other (in this case the discarded class), NorBERT2 seems to have a more balanced representation between the classes. It is the only model that is able to identify both classes to a certain degree, even if it still confuses many of the upheld claims as discarded claims. If we were to select a model that works

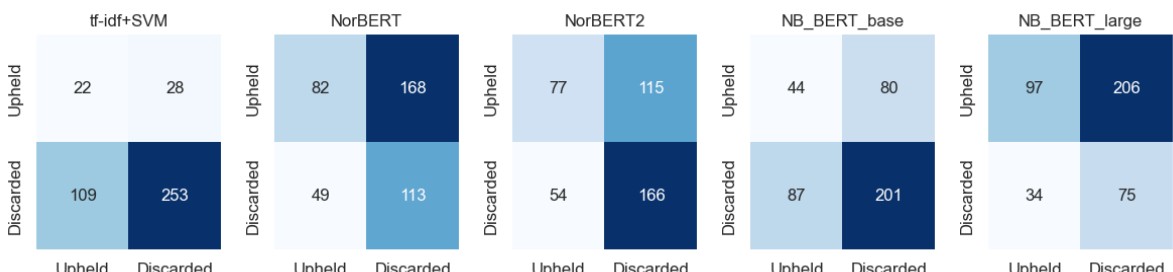

Figure 2: Confusion matrices of our models' predictions.

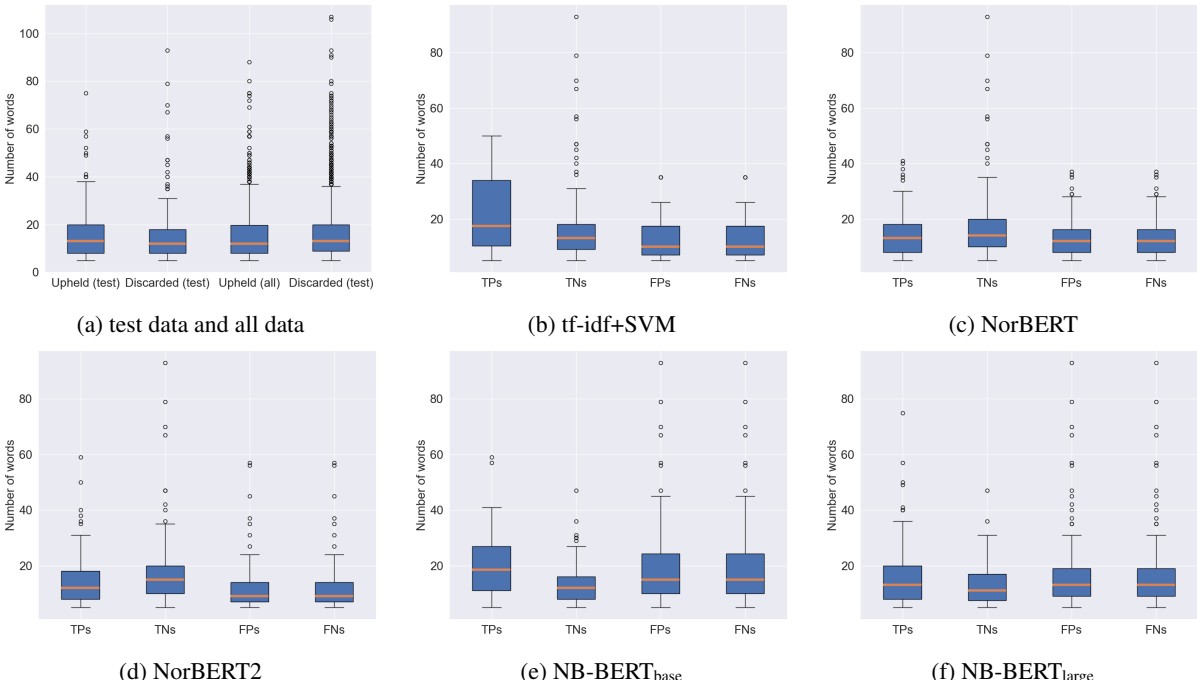

Figure 3: Distribution of number of words in claims across true and false predictions for the four Norwegian language models and the SVM baseline.

fairly good on both classes, NorBERT2 would be the natural choice.

Further analysis is conducted on the length of the claims with respect to the model predictions for true positives (TPs), true negatives (TNs), false positives (FPs), and false negatives (FNs). Figure 3 illustrates the box and whisker plots of the number of words in each of these groups. In Figure 3 (a), the number of words in the upheld and discarded class are shown for the test set and the whole data set. The length of the claims in the discarded class appears to be slightly larger than the upheld class. However, the quartiles and the median length are very close and thus length is not a significant discriminative feature. For the baseline model, length plays an important role in the model behaviour, though. The SVM model

correctly classifies the longer claims from the upheld class and the shorter claims from the discarded class. Among the LMs, NorBERT and NB-BERT_large are less sensitive to the length of the claims, as inferred from the similar statistics for true and false predictions. The figure also indicates that NorBERT2 suffers when predicting shorter claims, while NB-BERT_base deteriorates for longer claims from the discarded class.

We also looked into the sources of the incorrectly classified claims for different models. The five most frequent sources in the data set, namely, 'Politisk kvarter', 'Facebook', 'NRK', 'Dagbladet', and 'Debatten' are considered. The percentage of the claims with false predictions from each source are shown in Figure 4. One interesting observation is claims from 'Facebook' are

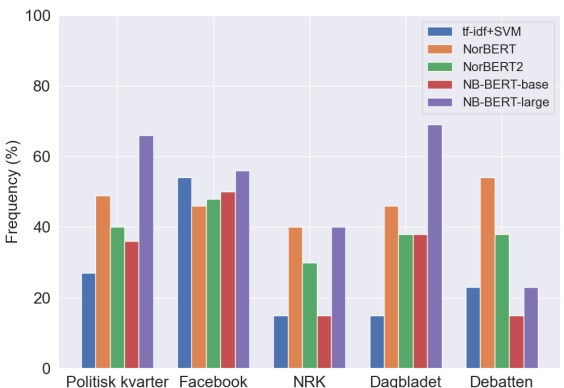

Figure 4: Percentage of the incorrectly classified claims from the five most frequent sources.

| Data Set Label | #Claims | Acc. |
|---|---|---|
| Discarded | 281 | 59.1% |
| Pre-checked and rejected | 37 | 56.8% |
| Pre-checking | 34 | 52.9% |
| Published | 30 | 56.7% |
| Suspended | 19 | 73.7% |
| Checking | 11 | 45.5% |

Table 4: Number of claims and accuracy in terms of original labels for the test set.

relatively difficult for all the models, while predicting the ones from 'NRK' seem to be more straightforward. This could be due to the differences in the writing styles in an official broadcasting organization and a social media platform. It is notable that the patterns for NorBERT and NorBERT2 are relatively similar across different sources. NB-BERT$_{base}$ and NB-BERT$_{large}$ are more sensitive to the source of the claims.

Finally, the predicted labels in the test set are analysed to see what percentage of each individual original label is correctly classified. We only focused on the NorBERT2 as it is the best model in terms of F1. Table 4 shows the number of claims in each category and the accuracy. The results are relatively comparable among the labels, which confirms the consistency of the mapping applied to convert the original labels to the binary labels. The two exceptions are Suspended and Checking class corresponding to the highest and the lowest accuracy, respectively.

## 5 Conclusion

In this work, we conduct a case study using Norwegian pre-trained LMs for the task of automated claim detection. Four existing Norwegian models in addition to an SVM baseline system are examined and compared using a claim detection data set that resembles a real world problem. The results show that language models outperform the baseline system. Different models can be selected for different purposes. If the overall performance is to be prioritized, the NorBERT2 model is the best performing. If the recall is the focus, then the biggest NB-BERT$_{large}$ model is to be selected.

Most of our observations can also be due to the differences between the LMs. The behaviour of our models can be due to model architecture, training procedures, and the datasets they were originally trained on. We also show how the length and the source of the claim plays a role in prediction patterns. We believe that there is more that can be uncovered from the behaviour of these models, and we plan to explore this in future works.

## Limitations

Our work does have some limitations that might have impacted the outputs of our models. For instance, the behaviour of the models might partly be due to the skewed distribution of classes in the dataset, where the discarded class is the majority class. Another limitation is publishing the data to reproduce the results and perhaps to conduct further analysis. Faktisk provided the data set to us to investigate automated fact-checking systems and publish the results. At the moment, we are not permitted to make the data set publicly available, as it is part of the organization's internal procedure. This might hopefully change in the future.

## Acknowledgements

We would like to express our deepest gratitude to Faktisk.no for sharing the data set with us for research purposes and for their insightful remarks on the fact-checking process.

This research was supported by industry partners and the Research Council of Norway with funding to MediaFutures: Research Centre for Responsible Media Technology and Innovation, through the Centres for Research-based Innovation scheme, project number 309339.

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
