# OpenReview forum: "Automated Claim Detection for Fact-checking: A Case Study using Norwegian Pre-trained Language Models"
_NoDaLiDa/2023/Conference — NoDaLiDa 2023_

### Official Review · Reviewer_yiXV · 2023-02-24
**Using classifiers based on pretrained BERT models for Norwegian in order to detect whether claims from social media need fact checking or are discarded**

**Rating:** 7
**Confidence:** 4

**Review:**

This paper relates to an important field, namely the fact-checking of claims made on social media. Fact-checking consists of three phases: claim detection, evidence retrieval, and claim verification. The current paper deals only with the first phase: claim detection.

Four BERT-based models are compared to a baseline SVM model. The BERT-based models generally have higher recall, but the baseline has higher precision. The experiments are generally performed well and the analysis is appropriate. The paper is also clearly and well written.

The following items would need clarification:

* The categories mentioned (displayed in Table 1) are not totally clear. What is the difference between a discarded claim (to be put in the discarded class) and a pre-checked and rejected claim (to be put in the upheld class)? Is this division into two classes a good one and the only reasonable one? Could there be examples of actual data to illustrate what the claims of the different categories are like? (This is the NoDaLiDa conference: you can put examples in Nordic languages, with English translations when necessary.)

* In Table 3 (and in the text related to it) it should be clarified whether the metrics are for the positive class only or both classes or what. The informed reader can figure it out, but please spell it out as well in the caption. What would be the result of a majority class baseline or a naïve baseline? Those figures could be useful as well, since the classes are skewed.

* In Figure 2, for the confusion matrices, please indicate which dimension is the desired and which one is the actual outcome. This can be figured out, but please make it explicit.

* Figure 4 is referred to as Figure 1 in the text.

In general, it would be nice to see more real data related to the task. What are the texts like? What are the potential claims like?


**Paper Type:**

Long paper

---

### Official Review · Reviewer_C1SE · 2023-03-08
**Decent first take on fact checking for Norwegian. Main contribution is dataset and initial results. It is unclear if dataset is actually distributed.**

**Rating:** 7
**Confidence:** 3

**Review:**

## Summary
This paper describes an initial attempt at modeling claim detection in the Norwegian language. Since there are no available datasets the paper considers a dataset of 4k samples originating from a Norwegian fact-checking organization. It is unclear if this data is publically available, or if it is distributed along the paper. The dataset contains several labels but casts the task into a binary classification task by conflating labels. The dataset is unbalanced, containing twice as many negative examples as positives. The paper documents the training of four Norwegian reimplementations of BERT and evaluates them with recall/precision/F1. The results are not overwhelming with the best models providing an F1-score of 0.47 and a recall of 0.740. It is unclear from the paper if this is a useful range of performance.

## Review
The paper documents a methodical approach to building a claim detection system for the Norwegian language. It is well-written and contributes to what I believe is reasonable evidence on a task that is so far untouched for Norwegian. It is not entirely clear to me if these systems have any practical use, but at the very least this paper provides a well-documented experiment that future work can build on. There are a few things I think would improve the paper:
- Clearly state whether the data will be available to others, or if it can be reproduced in future research. If not, the contribution of the paper is significantly weakened as there is no way to build on top of this work.
- On line 366 you mention that you actually cherry-pick a data split in favor of the SVM baseline. I strongly recommend not doing this as it is considered bad practice and imposes an unnecessary bias with no immediate benefit to the experiment. I don't think it invalidates the results for the BERT models, but it weakens the evidence.
- The results are aggregated over the stratified test set. Are there noticeable differences between the original labels?
- There is a very high variance between the results of models, suggesting that the models may be unstable. May it be worth considering a time-tested multilingual model like mBERT?

## minor
- on line 612 you reference figure 1. I believe you meant to reference figure 4

**Paper Type:**

Long paper

---

### Official Review · Reviewer_RVSH · 2023-03-10
**Classification with BERT and a linear layer**

**Rating:** 6
**Confidence:** 4

**Review:**

In this paper, the authors present an evaluation of contextualized LMs for Norwegian for classifying if a proposition should be flagged for human fact checking. The focus is classifying ‘check-worthiness’, not the factual content of the claim. The authors have two main claims they investigate: how well do Norwegian LMs perform, and which aspects do they struggle with.
The data comes from the fact-checking site faktisk.no, which looks like a great initiative. However, the authors provide little detail on the data source. A common critique of fact-checking is that someone has to fund the work, and that this greatly influences which and how claims are checked. The authors should state that some sources of claims in the data set comes from the news organizations funding the project. Even though the task isn’t classifying the fakeness of a claim, this limitation should be acknowledged and discussed, especially as Norway has Bokmål and Nynorsk. As an outside observer, I would like to know if there is a risk that “alternative” news sources use a less formal Norwegian, which is then picked up as suspicious by BERT. In this time of droves of bias papers, this could be easily tested and shown (perhaps qualitatively).
The pipeline is fairly straightforward. A contextualized language (four in total as tried separately) model projects shorter texts to a space, a linear classifier (linear layer) then separates this space into two output classes (compare to the IMDB polarity task). The reproducibility as very high, as the paper includes everything from pre-trained model versions to training parameters. The baseline is a tf-idf representation with a binary SVM-RBF classifier. This is a great baseline, though the sklearn implementation used in the paper does not compensate automatically for unbalanced classes (this is implemented as an extra option when instantiating the model, as far as I remember). This might explain why the baseline recall was so bad, while the precision was higher than for the more recent LMs.
The evaluation is well-made, showing that contextualized models are great, though tf-idf holds up surprisingly well in some situations. The contribution is not in surprising results, but that someone did a systematic evaluation. However, as the models are only fine-tuned, there would have been time for cross-validation instead of static train/validation/test splits. Even the largest model takes less than two minutes to train. Moving away from cross-validation in the recent decade has been motivated by the computational cost of training large models. This is not an issue here. Without cross-validation, the result becomes more of a proof-of-concept than an evaluation of the performance we could reasonably expect (which seems to be the authors’ intention).
It would be interesting to see how well the method work on newer data. I sometimes get the sense that fact-checked claims can be clustered in special political interests (e.g. immigration) or conspiracy theories (e.g. Covid-19 has some relation to 5G). I would find it interesting to test if the classifier is more likely finding known clusters, or if claims could be detected as check-worthy due to some shared features (e.g. structure of the argument). Perhaps showing the generalizability by testing claims on new issues that come after the end of the data period, perhaps on the Ukraine war, as it is sometimes said that the first casualty of war is the truth. I say this in light of the technical contribution not being very high and that they write that they focus on evaluation.
The second weak point (after not having cross-validation), is that the error analysis is over-relying on arguments from metrics like true positives, false negatives etc. There should be more analysis of the kind the authors write in the last paragraph of their discussion section, where they compare error rates for different news sources. I’d also find it very interesting to see what kinds of claims were hard to classify. Perhaps picking cherries and lemons among the claims, especially to a conference where most participants can read Norwegian. For example, how does a claim like “Norway has taken the least number of refugees in Europe” (from the data source) work with the models? Could it easily be manually changed to confuse the classifier (think adversarial examples)?
A layout point is that I’d prefer more explanatory captions to all tables and figures. If a reader wants a fast overview before reading the paper, trying to make the figures and tables “self-contained” is a good way of accommodating them.
The technical contribution is not very high, but the authors instead focus on evaluating some contextualized models for Norwegian on a specific task. Methodologically, I find not using cross-validation for evaluating model performance under changing training data a serious limitation to claiming anything about performance in a general case. Furthermore, as the error analysis does not go into specific examples, the authors fail to answer their stated research question “Which aspects of claim detection do these LMs still struggle with?”. Given these two objections, the paper is very well written. Hence, I recommend a weak accept.

**Paper Type:**

Long paper

---

### Decision · Program_Chairs · 2023-03-17

Accept